# Providing Food and Nutrition Services during the COVID-19 Surge at the Javits New York Medical Station

**DOI:** 10.3390/ijerph18147430

**Published:** 2021-07-12

**Authors:** Emily Sanchez, Amy R. Gelfand, Michael D. Perkins, Maia C. Tarnas, Ryan B. Simpson, Jarrod A. McGee, Elena N. Naumova

**Affiliations:** 1Friedman School of Nutrition Science and Policy, Tufts University, Boston, MA 02111, USA; Ryan.Simpson@tufts.edu (R.B.S.); Elena.Naumova@tufts.edu (E.N.N.); 2Army Medical Department Student Detachment, U.S. Army Medical Center of Excellence, Fort Sam Houston, TX 78234, USA; 3Bureau of Supplemental Food Programs, Division of Nutrition, New York State Department of Health, Albany, NY 12204, USA; arg315@nyu.edu (A.R.G.); mperkins377@gmail.com (M.D.P.); 4Community Research Initiative, Charlestown, Boston, MA 02111, USA; maia.tarnas@gmail.com; 51st Medical Brigade, 11th Field Hospital, Fort Hood, TX 76544, USA; jarrod.a.mcgee@gmail.com

**Keywords:** COVID-19, clinical nutrition, emergency response, field hospital, food service, medical records, New York City, US Army

## Abstract

Military field hospitals typically provide essential medical care in combat zones. In recent years, the United States (US) Army has deployed these facilities to assist domestic humanitarian emergency and natural disaster response efforts. As part of the nation’s whole-of-government approach to the coronavirus disease (COVID-19) pandemic, directed by the Federal Emergency Management Agency and the Department of Health and Human Services, during New York City’s (NYC) initial surge of COVID-19, from 26 March to 1 May 2020, the US Army erected the Javits New York Medical Station (JNYMS) field hospital to support the city’s overwhelmed healthcare system. The JNYMS tasked a nutrition operations team (NuOp) to provide patient meals and clinical nutrition evaluations to convalescent COVID-19 patients. However, few guidelines were available for conducting emergency nutrition and dietary response efforts prior to the field hospital’s opening. In this case study, we summarize the experiences of the NuOp at the JNYMS field hospital, to disseminate the best practices for future field hospital deployments. We then explain the challenges in service performance, due to information, personnel, supply, and equipment shortages. We conclude by describing the nutrition service protocols that have been implemented to overcome these challenges, including creating a standardized recordkeeping system for patient nutrition information, developing a meal tracking system to forecast meal requirements with food service contractors, and establishing a training and staffing model for military-to-civilian command transition. We highlight the need for a standardized humanitarian emergency nutrition service response framework and propose a Nutrition Response Toolkit for Humanitarian Crises, which offers low-cost, easily adaptable operational protocols for implementation in future field hospital deployments.

## 1. Introduction

Countries across the globe have erected field hospitals to support permanent medical facilities during the ongoing novel coronavirus disease (COVID-19) pandemic [1,2,3,4,5,6]. Often enacted by provincial or territorial health ministries, these facilities addressed the growing limitations of bed capacity, especially for non-critical care patients [4,5]. In Canada and Wales, field hospitals were erected in mobile tents, sports arenas, and vacated university campuses [4,5]. Even with increased bed capacity, the resource and personnel scarcities inhibited optimal medical treatment of infected and convalescent patients. This was especially true for mental health and nutrition services, which became high-priority medical services in the US Agency for International Development (USAID) 2021 global COVID-19 relief packages [7].

The US Army expeditionary combat support hospitals (CSH) and field hospitals typically provide essential medical services, including triage and emergency care, outpatient services, inpatient care, clinical laboratory and microbiology, blood banking, radiology, physical therapy, medical logistics, operational dental care, general and specialty surgery, nutrition care, behavioral health, and patient administration services, in combat zones [8,9,10]. These facilities can also be augmented by one or more specialty teams and military detachments, to further increase their capabilities and capacities [10]. Recently, however, the Federal Emergency Management Agency (FEMA) has deployed these medical facilities to assist in domestic disaster relief. In 2005, the 10th CSH mobilized to Louisiana following Hurricane Katrina [11], while in 2017 the 14th CSH mobilized to Puerto Rico after Hurricane Maria [12]. These successful operations have since informed Army planning and invited future implementation of field hospitals during domestic humanitarian crises, with the most recent being the COVID-19 pandemic. In fact, the Army Corps of Engineers and contracted private entities spent USD 660 million in 2020 on building field hospitals throughout the US [13]. This reflects a larger shift in US military operations to support domestic humanitarian emergency responses [14].

The response efforts have perhaps been the greatest in New York City (NYC), which was an epicenter of the US COVID-19 epidemic, after reporting ~230,000 cases in March–May 2020 [15,16]. The staggering volume of infected persons overwhelmed the city’s healthcare system, and left many facilities without sufficient personal protective equipment (PPE) and other supplies for treating patients [17,18]. To decompress the NYC outbreak, the US Army collaborated with public health officials to erect field hospitals that provided patient care that was unmet by local hospitals [19]. These sites, such as the Javits New York Medical Station (JNYMS), quarantined individuals with confirmed mild to moderate COVID-19, provided intensive care unit (ICU) capacity for patients in critical conditions, and discharged persons after recovering from the illness [20].

The JNYMS field hospital was also tasked with providing crucial and timely nutrition services and clinical nutrition assessments to patients. According to the United Nations, prior to the pandemic, nearly 690 million people were undernourished; COVID-19 caused another 270 million people to be in active hunger [21]. As a result, global programs, such as the Food and Agriculture Organization, the International Fund for Agricultural Development and the World Food Program, aimed to identify acute food insecurity hotspots, in order to provide food assistance [21,22,23]. Within the first month of this initial COVID-19 surge, food insecurity among NYC residents doubled, from approximately one million to approximately two million persons, so that nearly one in four New Yorkers lacked access to sufficient food [24]. Barriers to food access [25], a 3.1% increase in grocery costs [26], and soaring unemployment [27] further threatened increases in malnutrition among patients. Though unknown at the time, recent studies have since confirmed that malnutrition can increase the risk of ICU admission and mortality for COVID-19 patients [28,29,30,31]. The suspected mechanisms include expedited COVID-19 progression via a suboptimal immune response, due to an imbalance of protein and/or energy intake over time [28].

Despite mandates to perform emergency food and nutrition services, the JNYMS nutrition operations team (NuOp) lacked the operational protocols and formal guidelines for integrating nutrition assistance and services during deployment. The Army Techniques Publication 4-02.10 ‘Theater Hospitalization’ provides a doctrinal foundation for nutrition service operations in a deployed setting, and the Academy of Nutrition and Dietetics provides guidelines for conducting nutrition assessments. However, these tools were not easily adapted to emergency response settings, such as COVID-19 field hospitals, with civilian augmentation, and where PPE limitations required <5 min patient contact time to reduce clinician exposure. Furthermore, these protocols and guidelines did not extend to creating inpatient nutrition surveillance systems, managing food service contracts, forecasting meal deliveries, and augmenting military staffing models to include civilian nutrition personnel, in order to maintain an optimal operational performance.

In this case study, we describe the experiences of the JNYMS field hospital’s NuOp during deployment, from 26 March to 1 May 2020. We structure our findings as follows: First, we describe the organization of the JNYMS field hospital, composition of the NuOp, and the timeline of events during deployment. Next, we explain the operational mandate of the field hospital, as prescribed by the US Army, which included the following: (i) providing therapeutic meals; (ii) conducting inpatient medical nutrition therapy (MNT) and critical care support; and (iii) managing military-to-civilian staffing transitions. Lastly, we discuss the challenges faced when executing this mandate, and the solutions implemented to overcome these challenges. The NuOp staff experiences necessitate the creation of a standardized humanitarian emergency nutrition service response framework, offering low-cost, easily adaptable operational protocols for implementation in future field hospital deployments. We propose this framework, entitled the Nutrition Response Toolkit for Humanitarian Crises, with ready-to-use materials as necessary to conduct nutrition services in emergency response settings (Appendix A) [32].

## 2. Overview of JNYMS Structure, Staffing, and Timeline

The JNYMS was located at the Javits Convention Center in the Hudson Yards area of Lower Manhattan, NYC (Figure 1). The JNYMS consisted of seventeen 32-bed pods and two 48-bed intensive care units (640 total beds) with a maximum capacity of ~3000 beds. The hospital administrative staff established a command center two floors above the main hospital. This area acted as the command center for coordinating health services among the ~42 agencies involved in running the facility. The NuOp consisted of 13 registered dietitians (RDs) and 25 nutrition care specialists (enlisted soldiers knowns as 68Ms, pronounced “sixty-eight-mikes,”) from the US Army’s 44th and 1st Medical Brigades, US Public Health Service Rapid Deployment Team (USPHS RDF), the US Army Reserve Urban Augmentation Medical Task Force (UAMTF) 804-2 and 338-1, and the New York State Department of Health’s Division of Nutrition (NYS DOH).

Army RDs are officers who are trained to support operational brigades, through the delivery of comprehensive performance nutrition focused on foundational health (i.e., chronic disease prevention and immune system enhancement), and environmental and task-specific performance (i.e., event fueling and post-event recovery, body composition, mental function, and arduous environment preparedness) [33]. Additionally, Army RDs provide nutrition support to service members through Defense Health Agency medical treatment facilities (MTFs), and humanitarian assistance in the form of nutrition aid and resources [33]. The 68Ms are enlisted soldiers who serve as extensions of an RD’s workforce. The 68Ms are trained to conduct health promotion events, inpatient malnutrition screening, diet education, and food service operations for therapeutic meals within field hospitals and treatment facilities [33].

Within a field hospital, Army RDs and 68Ms serve as the primary hospital staff, who are responsible for the nutrition care operations [10]. Army RDs are responsible for managing the medical food preparation, service system and staff food preparation services, coordinating procurement and receipt of safe, wholesome food items/rations for patients and staff and medical diet supplements for patients, and assisting physicians by providing patient nutrition assessments, nutrition therapy recommendations and medical nutrition therapy [10]. Whereas, the 68Ms assist in the nutrition screening and assessment of individual patients, as well as preparing, serving and delivering modified and regular food items in the management of the nutrition needs of patients, and monitoring and maintaining the inventory control of subsistence, supply and equipment [10].

During the approximately one-month deployment, the NuOp experienced rapidly changing hospital staffing arrangements (Figure 2). The NuOp staff, consisting of the US Army’s 44th and 1st Medical Brigades and USPHS RDF, arrived to the JNYMS on 28 March, only two days before it opened, to care for the non-COVID-19 patients. At this time, the length of deployment was unknown, though the PPE and other medical supply preparations were aimed to support field hospital operations for 30–60 days. To begin preparations, the NuOp staff and tasks were designated to either ‘Foodservice Operations’ or ‘Clinical Nutrition Operations.’ Moreover, prior to admitting patients on 31 March, the NuOp confirmed food contract delivery schedules, obtained access to non-electronic patient census tracker forms, and established protocols for malnutrition screening.

On 3 April, the local healthcare providers requested that the JNYMS accept COVID-19-positive patients who required moderate-to-high levels of oxygenation and care. This request was in response to the increased incidence of NYC cases and deaths, and the decreased bed capacity citywide (Figure 1b–d, respectively). On 5 April, the JNYMS began admitting COVID-19-positive patients, and to protect the health of foodservice workers, all patient meal preparation was moved offsite. In response to the growing patient census, UAMTF RDs were deployed on 11 April, to join the NuOp. Increased personnel were critical for expanding the food service operations, in order to accommodate patient dietary restrictions and increase nutrition consultations among those patients who were at-risk for malnutrition.

After successfully managing the surge of patients, the NuOp began preparing for military-to-civilian staffing transitions on 16 April. This transition included the refinement of patient diet order forms and meal forecasting calculators for use by incoming civilian personnel. On 23 April, the NYS DOH RDs arrived to the JNYMS and joined the NuOp staff, who trained all the incoming personnel by 25 April. On 28 April, military personnel transitioned all the food and nutrition service operations to the NYS DOH RDs. The facility officially closed on 1 May. The temporal dynamics in patient census, number of NuOp RDs, and the total meals served per day, from 27 March to 1 May 2020, are shown in Figure 1f–h, respectively.

## 3. JNYMS NuOp Mandates

According to US Army doctrine, in a deployed setting, nutrition care operations are responsible for meal preparation and service to patients, MNT, dietetic planning, patient education, and theater health promotion [10]. At the JNYMS, this translated into the following three core mandates for the NuOp: (i) provide patient therapeutic meals, (ii) perform inpatient MNT and critical care support, and (iii) manage the JNYMS staffing transition from military-to-civilian personnel. Each mandate was carried out by members of the NuOp. The tasks under each mandate, often conducted simultaneously by all the staff members, followed a carefully constructed protocol to meet the changing needs and demands of COVID-19 patients (Figure 3). The NuOp services were initiated once the patient was admitted to the JNYMS and underwent malnutrition screening. In this flowchart, key decision-making steps and treatment protocols for food service and clinical nutrition operations are visualized, using shapes and colors to reflect the start, process, and decision steps, and which personnel performed those steps, respectively. The NuOp services contributed to the patient’s medical care while admitted, optimally concluding with a healthy discharge from the JNYMS.

### 3.1. Food Service Operations

Prior to NuOp arrival, the JNYMS hospital administrators established a food contract to provide patient meals. The contract provisions included three cold meals per day per patient, hot beverages, and floor nourishments, such as fruit cups, milk, and chips. Five therapeutic meals were offered on a four-day cycle, which included the following: (i) a ‘core four’ meal, appropriate for regular, consistent carbohydrate, heart healthy, and low-sodium diets; (ii) a renal diet; (iii) a gastrointestinal soft diet; (iv) a National Dysphagia Diet Level 2 and Level 3: ground/chopped; and (v) a National Dysphagia Diet Level 1: puree. To ensure patient safety, the National Dysphagia Diet Level 2 and Level: ground/chopped meals were prepared with the more restrictive texture modification of ground consistency. The full liquid and clear liquid diets were assembled using the available floor nourishments, as needed. By 17 April, hot soups and hot lunches were made available to the patients (Table 1).

Critically ill patients received a limited enteral nutrition formulary, consisting of a standard high-protein, polymeric enteral formula to sustain the patients’ protein levels before their transfer to an NYC permanent hospital facility. This formulary was initiated per the American Society of Parenteral and Enteral Nutrition guidelines to patients previously intubated in the ICU [38]. Patients requiring parenteral nutrition similarly received a premixed parenteral nutrition solution before being transferred. Stable critical care patients and those requiring chronic nutritional support received a fiber-containing enteral formula.

Food service contractors prepared and packaged the patient meals offsite. Staff from the USPHS RDF monitored the food temperatures prior to delivery. All of the meals were transported ~13 miles daily and stored in refrigerated locations to ensure cold chain management at ≤41 °F (5 °C). The NuOp created a meal staging area within the hospital, to prepare the meal delivery carts and oversee the inventory of meals, snacks, and oral nutrition supplements. The 68Ms monitored their assigned pods for new admissions, room transfers, or discharges within 1 h of meal delivery. New admissions were interviewed for food allergies, dietary preferences, meal texture modifications, and religious/cultural considerations. Diet forms were relayed to the RDs, who ensured appropriate meal or nourishment substitutions were available to that patient.

During meal delivery, the 68Ms verified diet orders, assembled food delivery carts, and delivered patient meals. Concurrently, the RDs collaborated with the patient administration personnel to monitor new admissions, room transfers, and discharges. This oversight minimized the errors or omissions of meal provisions for patients undergoing turnover or transfer during this servicing period. The RDs also received verbal requests for new or modified diet orders, oral nutrition supplements, and/or consultation by medical staff.

### 3.2. Clinical Nutrition Operations

RDs assessed and monitored patients in the ICU for nutrition risk and calculated nutrient requirements to determine the appropriate nutrition support therapy. Nutrition support therapy included the provision of enteral (tube-fed) or parenteral (intravenously fed) nutrients to treat or prevent malnutrition [39]. The NuOp had 12 enteral feeding pumps and additional gravity feeding bags to overcome pump shortages. The NuOp built an oral nutrition supplement and enteral nutrition formulary, based on the food supplies requested in the JNYMS food contract. RDs also consulted with the hospital’s pharmacy to identify parenteral nutrition needs (i.e., types of and quantities available of premixed nutrition solutions for intravenous administration) and procurement channels, as these supplies were not available through the food service contract.

The clinical nutrition operations included ‘critical care’ and ‘floor nutrition care’ teams. Each team consisted of 2–3 dedicated RDs and 7–10 68Ms, who performed clinical tasks in addition to food service responsibilities. The 68Ms screened patients for malnutrition, using a questionnaire developed with clinical judgement that evaluated the patient’s appetite, diet, weight history, and chewing and swallowing difficulties. Patients were considered at-risk for malnutrition if they reported fair or poor appetite and recent unintentional weight loss. At-risk patients received additional nutrition counsel, and the RDs documented patient care and recommendations in the patient’s medical chart, using a modified version of the Nutrition Care Process note-writing style, ADIME (assessment, diagnosis, intervention, monitoring, and evaluation) [40,41]. The recommendations from these assessments were relayed to the patient’s primary provider, in order to optimize the continuity of care.

### 3.3. Sustaining Nutrition Operations

The military-to-civilian transition of nutrition operations required careful planning of staffing, equipment, and supplies, to ensure the continuity of care. The NuOp personnel needs were adapted from the existing military staffing models, to ensure the facility remained fully operational despite the fewer personnel. To manage the transition, the NYS DOH leadership recruited NYS DOH RDs with multi-operational experience in the areas of food service, critical care, and acute care.

NYS DOH RDs were screened for operation-specific considerations, including their willingness to work in a COVID-19-positive facility and possible safety concerns for the recruit’s personal or family health. Three NYS DOH RDs assisted in the operational handoff and made recommendations for additional staffing needs. All of the NYS DOH RDs completed a comprehensive two-day training with military RDs. These trainings explained the organization of the JYNMS, operational mandate for the NuOp, and the coordinated protocol for simultaneously conducting food services and performing clinical nutrition evaluations. The NuOp identified an acting civilian clinical nutrition manager to provide daily updates to multi-agency hospital administrators.

## 4. Challenges and Solutions

While deployed, the JNYMS became the largest Department of Defense inpatient facility and the 13th largest hospital facility in NYC. In that time, the NuOp served >15,000 patient meals, with no patient reports of inappropriate diet orders or food allergy-insensitive meals. The 68Ms screened all of the admitted patients (~1100) for malnutrition risk, within 24 h of patient admission. A total of 130 patients (~12%) screened at-risk for malnutrition and were referred to an RD for individualized MNT. The RDs assessed an additional 100 patients following consultations from other JNYMS medical staff.

Despite its successes, the NuOp experienced various challenges, both in overcoming medical supply and personnel shortages and when transitioning the JNYMS from a non-COVID-19 to a COVID-19 field hospital, and from military- to civilian-led operations. These challenges were ultimately overcome due to the NuOp’s adaptation of existing food and nutrition resources for the fast-paced and ever-changing hospital environment caused by COVID-19. Below, we discuss the challenges faced by the NuOp staff and explain the solutions taken to establish standardized protocols to overcome these challenges.

### 4.1. Establishing Food Contracts as Part of a Preparedness Protocol

Hospital administrative personnel typically establish food contracts without consultation from nutrition professionals. The failure to incorporate nutrition operations staff into this contract development impedes efficient coordination and delivery of diet orders. At the JNYMS, medical personnel frequently solicited the RDs’ guidance when planning meals to best support medical prognoses, adjusting menu options for dietary restrictions, and building enteral formulary. However, no publicly available recommendations exist for guiding RDs in selecting menu options when establishing food contracts in emergency settings.

Fortunately, the NuOp RDs had ample prior experience working on food contracts in a humanitarian emergency response environment. To effectively coordinate the meal needs with the contractors, the RDs created menu templates that were adaptable for the variety of food supplies available. This adaptability ensured consistent accessibility to all menu options, irrespective of resource scarcities, and provided alternative meals that were suitable for a diverse array of dietary restrictions and formulary needs. In future field hospital deployments, we recommend incorporating RDs and other nutrition professionals in food contract coordination, especially in emergency response settings involving meal forecasting and delivery.

### 4.2. Need for an Electronic Medical Records (EMR) System to Ensure Continuity of Care

Several studies highlight the importance of an electronic medical record (EMR) system in disaster-related medical responses, as they improve the resilience of health services and the continuity of care [42,43,44]. However, the JNYMS hospital administrators were unable to establish an EMR that worked across all military and civilian entities involved while the field hospital was open. The lack of a universal EMR impeded the harmonization of nutrition and medical records between the hospital personnel. This both reduced the interoperability and continuity of care between the nutrition and medical branches, and delayed food service operations, including collecting, recording, and administering patient diet orders. More importantly, the lack of an EMR increased the difficulty of collecting and monitoring patient food allergy and dietary requirement information. This, in combination with various biomedical information forms (e.g., fluid intake and output, vital signs, biochemical and anthropometric data), required the RDs to spend large amounts of time reviewing paper medical charts, which impeded the efficiency of care.

To overcome this challenge, the NuOp developed a standardized data collection form, to acquire information on patients’ diet orders and food allergies (Table 2).

This form allowed for knowledge sharing between medical and nutritional personnel, by harmonizing the information on dietary supplements, food preferences, diet texture/consistency, and religious/cultural observances with physician treatment, prescriptions, patient admission status, room location or transfers, and hospital discharges. Furthermore, record reconciliation improved the coordination of meal delivery between the food contractors, 68Ms, and RDs, and acted as a safeguard to ensure appropriate and allergy-friendly diet orders. Because the form easily integrates with standardized patient bed rosters and medical data, we recommend digitizing this form for future emergency responses using EMRs.

### 4.3. Implementing Communication Tools for COVID-19 Facility Management and Food Service

The transition to a COVID-19-positive facility challenged the NuOp’s capacity to perform food service responsibilities, especially with limited PPE. Concern over the personal safety of the contracted food service employees forced meal preparation to relocate to an off-site facility. This delayed implementing changes to therapeutic meal offerings, as modifications to the diet orders were completed only after receiving the delivered meals. Furthermore, hospital security measures restricted access to the elevator that was originally used to expedite the diet orders from the meal preparation stations to patient bedsides. This required the meal staging area inside the JNYMS to relocate near an unrestricted elevator with limited storage, refrigeration, and warming capabilities. This relocation in meal preparation increased the amount of time required for food delivery, and introduced food safety and spoilage concerns.

To overcome this challenge, the NuOp created a digital, cloud-based diet order tracker, which was used to record the timing and quantity of the delivered meals. This tracker carefully integrated with the diet order and food allergy information, to serve as a monitoring tool for successful and accurate meal delivery (Table 3).

As shown, this tool records the number of persons served for breakfast, lunch, and dinner, according to each diet order. After each week, the total meals served provided an inventory estimate of the number of meals needed for the next week. This inventory assessment helped maintain adequate forecasting of meals with food contractors. Additionally, this tool helped the NuOp to maintain proper food safety assurance and meal delivery coverage, in accordance with the USPHS Food Safety regulations.

### 4.4. Adapting Malnutrition Screening Tools for Safe and Efficient Medical Nutrition Therapy

Nutrition screening was essential in identifying those at-risk for malnutrition, expediting a nutrition assessment, and providing appropriate MNT. Prior to the JNYMS becoming a COVID-19 facility, the NuOp considered using a common validated nutrition screener, the Mini Nutritional Assessment (MNA^®^) [45,46], to examine malnutrition risk. The MNA^®^ has six questions that ask about weight loss status in the last three months, using a Likert scale. The scores range from 12 to 14 (normal nutritional status), to 0 to 7 (malnourished), and are tallied from all questions to determine a patient’s nutrition status of either normal (≥12), at-risk (8–11), or malnourished (≤7) [47]. Another common screener that was considered was the subjective global assessment (SGA), which uses a quantitative approach by calculating percent weight loss over a specified time interval and a handgrip strength device [48].

Though both screening tools are validated and well known among the nutrition and dietitian community, neither screener could be used at the JNYMS because of the need for a substantially reduced exposure time (<5 min) between the 68 Ms and patients. Airborne transmission of COVID-19, in combination with a mandate to conserve the use of PPE, forced the NuOp staff to conduct nutrition screening while employing social distancing, face shields, and physical barriers. This impeded questionnaire completion and prevented the use of more extensive nutrition screening tools (such as the MNA^®^) that require more time to complete. Furthermore, social distancing and a lack of readily available medical equipment inhibited the collection of standard anthropometric measurements, such as weight or handgrip strength, or translation of anthropometric information to non-English speakers. Instead, the NuOp adapted the existing screeners to quickly identify malnutrition risk while minimizing exposure time.

Using the MNA^®^ for guidance, the NuOp developed a screener consisting of three questions with simplified language, to cater to all English language fluency levels (Table 4). Malnutrition risk levels were retained as a normal status if the patient reported a good appetite, denied recent weight loss, or reported recent intentional weight loss, and denied chewing or swallowing difficulties. Patients that reported fair or poor appetite (≤50% of normal intake) and recent unintentional weight loss were considered at-risk for malnutrition. By using a lower threshold for malnourishment than typical surveys, the NuOp staff attempted to reduce errors in clinical screening. Additionally, the screener included questions regarding the patient’s ability to chew or swallow without difficulty, cultural and religious food preferences, and food allergies. This information is typically collected during admission via a nurse interview, and is not a part of common validated nutrition screeners. Instead, this adapted screener provided immediate access to potential food safety concerns in the absence of an EMR. Furthermore, the 68Ms could easily conduct this screening without clinical nutrition expertise and relay the results to RDs, for more rapid, targeted in-depth nutrition evaluations and interventions.

### 4.5. Modifying Nutrition-Focused Physical Examinations (NFPE) for Clinician Health Protection

RDs play a critical role in diagnosing malnutrition through the assessment of the following criteria: (1) insufficient energy intake; (2) weight loss, (3) loss of muscle mass; (4) loss of subcutaneous fat; (5) localized or generalized fluid accumulation; and (6) decreased functional status [49]. A patient must meet a least two of these criteria to recommend a diagnosis of malnutrition. The NFPE is one tool that allows the RD to assess three key criteria, i.e., loss of muscle mass, loss of subcutaneous fat, and localized or generalized fluid accumulation. Included in the standard of practice for RDs starting in 2012, the NFPE is a full body examination that is intended to evaluate physical appearance and function, to assist in determining nutritional status [50]. The NFPE can also be used to uncover signs of nutrient deficiencies and toxicities. However, the social distancing protocols at the JNYMS limited hands-on interactions between the NuOp RDs and patients, to observe the physical manifestations of malnutrition and collect basic anthropometric measurements. Yet, limited nursing staff and manual patient records led to inconsistencies in the collection and dissemination of anthropometric records between nutrition and the medical staff. Furthermore, some NuOp RDs had not recently practiced clinical dietetics, leading to a lack of familiarization of this assessment tool, and, thus, underutilization when assessing for malnutrition.

To overcome these challenges and successfully identify patients that are at-risk for malnutrition, the NuOp RDs conducted individualized nutrition assessments, based on <5-min visual inspections. During the patient interviews, the RDs assessed the patient for a loss of muscle mass and subcutaneous fat, through visual inspection of the patient’s orbital fat pads, temporal, clavicle and dorsal hand muscles. The RDs also collected in-depth food and nutrition histories, with careful attention to the food items that are most and least comfortable to consume. Lastly, the RDs requested additional biomarker tests and anthropometric measurements when the patients presented with clear micronutrient or macronutrient deficiencies. Though limited compared to a full NFPE, this adapted protocol provided a resource to optimize nutrition evaluations under the constraints of limited patient interaction time and COVID-19 safety precautions.

### 4.6. Creating Recruitment and Staffing Models for Continuity of Services

When preparing military-to-civilian transition of RD staffing, the NuOp considered existing military field hospital staffing models, as well as staffing models that are traditionally applied in acute care settings [10,51,52,53,54]. However, these models are not directly transferable to military field hospitals with civilian augmentations, such as the JNYMS, which required longer times for completing patient assessments due to the absence of an EMR, donning and doffing full PPE, locating patients amid transfers, coordinating care with medical and nutritional personnel, documenting MNT recommendations, and balancing the completion of clinical and food service roles. These factors resulted in limited RD capacity to provide care to more than 3–4 patients per day.

As a result, the NuOp developed a revised staffing model (Table 5) for field hospitals, where teams of 4–5 RDs and 10–20 68Ms could provide care to ~500 patients. This model encouraged the hiring of civilian alternatives to the 68Ms, i.e., Nutrition and Dietetics Technicians, Registered (NDTRs) [51]. NDTRs are nationally credential, “…educated and trained at the technical level of nutrition and dietetics practice for the delivery of safe, culturally competent, quality food and nutrition services”, and may work to assist the RD in providing MNT in direct patient settings [55]. Since most NDTRs practice inpatient acute-care, long-term care facilities or social services organizations, a hiring agency would have been needed to recruit them appropriately [56,57]. Leadership should consult with staffing agencies who work closely with healthcare and hospitals, to recruit dietetics staff with the appropriate clinical skills and credentials. Additional companies have developed over the years, specializing in dietetics staffing, and should also be consulted with in order to provide the best staffing for disaster response efforts. Whenever possible, the clinical nutrition team of emergency response efforts should include NDTRs to work alongside the RD. The NuOp recruitment strategies pivoted to target RDs who are currently practicing in a clinical setting, or who had recent clinical experience (preferably within the past two years).

The NuOp also developed a comprehensive two-day training resource to orient civilian RDs and offer appropriate training on food service management, clinical nutrition, and nutrition support operations, for RDs who lacked recent clinical experience (Table 6). These trainings explained the organization of the JYNMS, operational mandate for nutrition services, and coordinated a protocol for simultaneously managing food services and performing clinical nutrition evaluations.

## 5. Discussion: A Call to Action

In this case report, we present the challenges faced by the JNYMS NuOp staff, and the steps taken to adapt the existing nutrition guidelines and protocols for emergency response settings. Although the literature is limited, similar challenges are being reported by health professionals across the world. For instance, Cermonesi et al. [58] reported on the preparation and service of meals aboard a ferry ship transformed into a hospital for COVID-19 patients in Italy. The staff gave particular attention to the patients’ religion and culture as well as clinical factors when preparing meals, and then donned all necessary PPE for safe meal delivery. Meanwhile, in France, Thibault et al. [59] summarized feedback from nutrition professionals treating COVID-19 patients during the early weeks of the pandemic. Ten challenges to clinical nutrition practice were emphasized, which include the restriction of COVID-19 hospital wards to healthcare teams that are deemed indispensable or dedicated, and did not always include an RD. With limited patient access, Thibault et al. reported that the RDs relied more so on teleconsultations, which may not always be feasible in a field setting. Additionally, the RDs identified similar barriers to screening and diagnosing malnutrition, due to a lack of scales, a handgrip dynamometer, and the usual tools for measuring arm or calf circumference [59]. These collective experiences highlight the lack of emergency response tools and guidelines for providing nutrition care to field hospital patients during humanitarian emergencies. These challenges necessitate the creation of a standardized nutrition emergency response toolkit for implementation in future field hospital deployments.

We have begun to compile such resources by creating a Nutrition Response Toolkit for Humanitarian Crises (see Appendix A), which provides tools that are designed to assist nutrition personnel in conducting nutrition services in a variety of emergency settings, with or without an operational EMR [32]. These tools can help reconcile medical and nutrition records, harmonize patient information, and improve the continuity of care between medical and nutrition branches. In the future, we hope this toolkit can address the challenges experienced by other RDs and nutrition personnel, by providing the following: (i) staffing models/recommendations for field environments operated by military and/or civilian entities; (ii) therapeutic meal templates that account for limitations in time, personnel, and medical supplies; and (iii) adaptable templates for diet order monitoring, malnutrition screening, and individualized MNT documentation. The presented material, including the developed flowcharts, also strengthen the theoretical foundations for establishing effective response policies and procedures.

While not exhaustive, our case report is one of the first public reports documenting the experience in setting up a food and nutrition unit during the pandemic. This study adds to a growing body of literature on the best practices disseminated by medical staff assisting metropolitan field hospitals worldwide [60,61]. For example, staff from the Baltimore Convention Center Field Hospital provide a four-step framework for rapid response when managing decompensating COVID-19 patients [60]. This protocol established a unified communication system for standardizing emergency responses and stabilizing rapidly decompensating patients with minimal personalized protective equipment (PPE) [60]. Alternatively, Smith et al. established a triage rehabilitation care framework for acute care patients, whose disease-driven impairments result in functional deficits [61]. This framework consists of resources for screening patient deficit severity with minimal patient–professional contact time, establishing a patient treatment plan for those with high, moderate, and no rehabilitation needs, and suggesting a staffing model for administering services with limited hospital personnel [61]. We hope that our case study will encourage further review of theoretical approaches and broad concepts, such as the theory of change [62] and implementation science [63]. We invite the reporting and dissemination of other medical staff experiences, both within and outside of nutrition services, to curate the best practices for future field hospital deployments. Future studies could include rigorous testing of the proposed toolkit under controlled scenarios of simulation exercises.

As the COVID-19 pandemic continues, we call upon the broader research and practice community, including the Academy of Nutrition and Dietetics, to develop guidelines for nutrition services in field hospital and temporary medical facilities that are responding to humanitarian crises. These guidelines must target training competencies in managing and implementing food contracts, and utilizing the NFPE, which are becoming vital skills in response to COVID-19. Once adopted and developed by the Academy of Nutrition and Dietetics, this toolkit could be available to the ~107,000 RDs and ~5000 NDTRs registered in the United States alone [64]. Additional training opportunities can ensure more dynamic relationships between food contractors and nutrition professionals, to coordinate food service operations in future disaster relief efforts. These trainings will also help establish a firm presence of nutrition services in critical medical care, and will advocate for the integration of nutrition personnel in the planning of health operations before future emergency health service deployments.

## 6. Conclusions

The operations at the JNYMS and within the NuOp largely benefited from the whole-of-government approach, in which FEMA authorized the mobilization of military service members to respond and provide medical and infrastructural resources (e.g., health care professionals, food access and storage, electricity) to the COVID-19 relief efforts. Given these circumstances, not all of the specified results, experiences, and challenges in this paper may be applicable to other humanitarian response efforts. However, the lessons learned from this experience, and the resulting toolkit, may be further examined, tested, and adapted, in whole or selectively, for other emergency responses. Since field hospitals require foundational components, such as staffing and patient tracking, these experience-derived operational recommendations and tools may be helpful in guiding the establishment of field hospitals across levels of resource access. They may also complement the United Nations’ policy recommendations that the food security and nutrition response to COVID-19 includes a focus on directing attention to areas in need of the most life-saving assistance, improving surveillance, and designating food and nutrition services.

## Figures and Tables

**Figure 1 ijerph-18-07430-f001:**
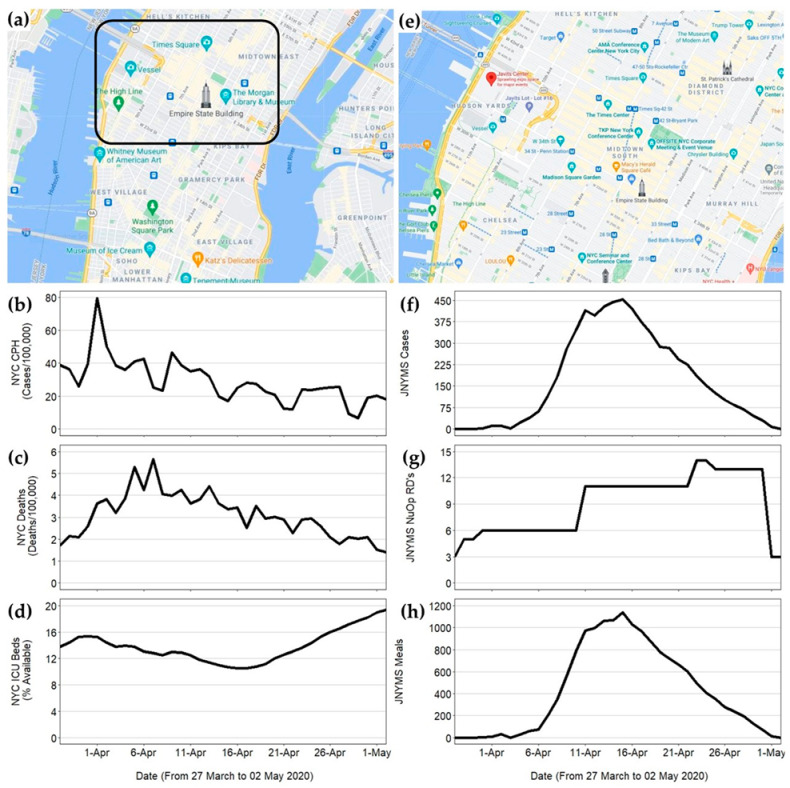
Maps of New York City (NYC) and the Javits New York Medical Station (JNYMS) location (panels (**a**) and (**e**), respectively) and multi-panel time series plots for NYC’s. (**b**) COVID-19 cases per 100,000 persons, (**c**) deaths per 100,000 persons, and (**d**) and hospital bed availability, as well as JNYMS’ (**f**) patient census, (**g**) number of NuOp RDs, and (**h**) total meals served per day from 27 March to 1 May 2020 [34,35,36].

**Figure 2 ijerph-18-07430-f002:**
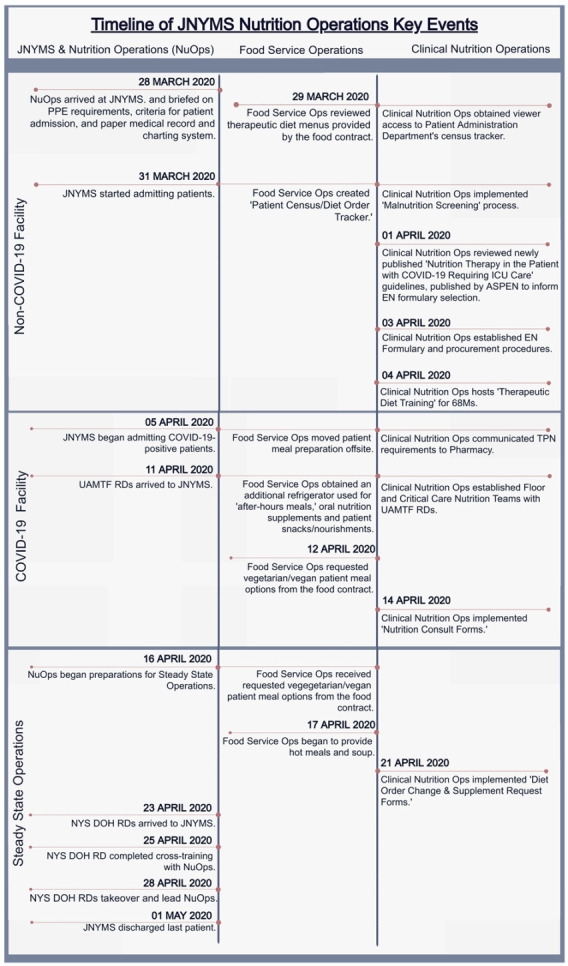
A timeline of nutrition operations events occurring at the Javits New York Medical Station (JNYMS) field hospital in New York City. This timeline includes JNYMS and NuOp personnel operations, food service operations, and clinical nutrition operations from NuOp personnel arrival on 28 March through the JNYMS discharging its last patient on 1 May.

**Figure 3 ijerph-18-07430-f003:**
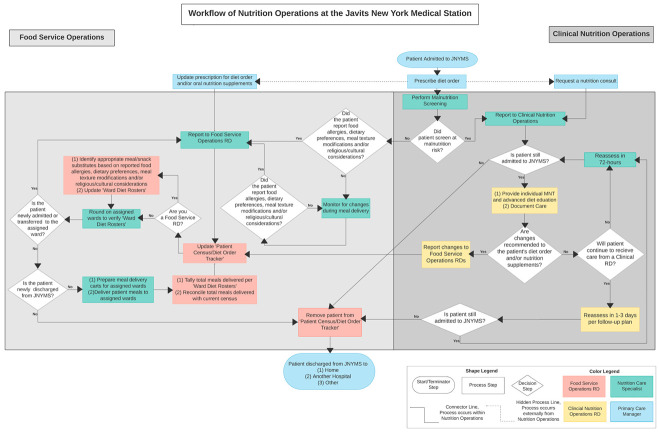
A process-based diagram demonstrating the workflow of food service and clinical nutrition operations at the Javits New York Medical Station performed from 28 April to 1 May 2020.

**Table 1 ijerph-18-07430-t001:** Example patient menu for the ‘core four’ meal option with hot items. The ‘core four’ meal was the most commonly prescribed therapeutic meal among Javits New York Medical Station patients from 28 April to 1 May 2020.

Meal	Example Food Items
Breakfast	Cheerios^®^, light vanilla yogurt, string cheese, skim milk, apple slices
Lunch	Beef pot roast with gravy, garlic mashed potatoes, steamed carrots, unsweetened applesauce, no-sugar-added chocolate pudding
Dinner	Tomato soup, unsalted crackers, turkey and Swiss sandwich, fruit cup, sugar-free cookie
Daily average nutrition content = 1540 kcal (85 g protein or 22% of kcals) *	

* Patient meals were often supplemented with floor nourishments and snacks as needed in order to meet the 2000–2500 calories per day and 75–100 g of protein per day recommendation for COVID-19 patients [37].

**Table 2 ijerph-18-07430-t002:** An example of the ‘Diet Roster’ form, designed to quickly identify patients’ diet orders, food allergies and oral nutrition supplement requirements during meal delivery. Nutrition personnel would be expected to check, circle, or cross out the meals received by each patient once that meal had been administered on that day.

		Diet Roster							
**Date**		1 January 2020		**Pod** #			3		
Line	Bed/ Room	Patient Name (Last Name, First Name, Middle Initial)	eFIND# *	Diet Order	Food Allergies	Oral Nutrition Supplements	Meal Delivered		
1	10	Doe, John	XXXXX	Renal	None	None	B	L	D
2	12	Smith, Jane	YYYYY	Core4	Nuts	Glucerna with breakfast	B	L	D
3	…	…	…	…	…	…	B	L	D
Provide this form to the food service dietitian at the end of meal delivery. Discuss changes to ward roster, barriers to delivering patient meals, etc.				Name of 68M assigned to Ward					
				68M’s name					

* The ‘eFIND#’ at the JNYMS is synonymous with a patient’s medical record number.

**Table 3 ijerph-18-07430-t003:** An example of the ‘Patient Census and Diet Order Tracker,’ designed to facilitate meal forecasting in a disaster response setting. This is done by recording the number of patients consuming different types of meals over pre-determined mealtimes in a field hospital. Within an hour of meal delivery, the tracker was cross-verified with ‘Diet Rosters’ and updated to communicate pre-service meal requirements to food contract staff. During meal delivery, the tracker was routinely updated as 68Ms completed their meal deliveries and/or as patients were admitted or discharged from JNYMS. Lastly, within 30 min of final meal delivery, the tracker was updated to reflect total meal served. Daily and weekly tallies of delivered meals were used to estimate the popularity of food supplies in the patient population. This tracker also served as a quality control tool in which the total number of meals delivered per meal period were matched to the current patient census, and this ensured every patient received a meal.

Item	5 January, Sunday			6 January, Monday		
	Breakfast	Lunch	Dinner	Breakfast	Lunch	Dinner
Meals Served	329	332	339	361	353	347
Total Census	329	319	330	343	333	340
Diet Orders						
Core4 *	300	297	306	320	315	316
Renal	11	5	0	0	1	1
Gi Soft	3	7	10	7	6	6
Ndd2&3	0	0	0	4	4	5
Pureed	0	0	0	0	0	1
Full Liquid	0	0	0	0	0	0
Clear Liquid	0	0	0	0	0	0
Npo	2	0	0	0	0	0
Kosher	6	8	9	11	7	9
Dairy-Free	7	15	14	19	20	9
Vegetarian/Vegan	0	0	0	0	0	0
Gluten Free	0	0	0	0	0	0
Nutrition Support	0	0	0	0	0	0

* Core four suitable for regular, consistent carbohydrate, heart healthy, and low-sodium diet orders.

**Table 4 ijerph-18-07430-t004:** An example ‘Nutrition Screening Form,’ designed to collect nutrition-related information on all patients simply and quickly.

Nutrition Screening Form																				
Patient Name (Last Name, First Name, Middle Initial)			Doe, John S.				Date								1 January 2020					
							eFIND #								XXXXX					
POD #			3				Room #								20					
Appetite History (mark ‘X’ where applies)							Weight Loss History (mark ‘X’ where applies)													
Good (75–100% of meals)							Recent weight loss?							Reason for weight loss?						
Fair (50–75% of meals)					X		Yes				X			Intentional						
Poor (<50% of meals)							No							Unintentional					X	
Chewing/Swallowing Difficulties? (mark ‘X’ where applies)							Cultural/Religious Food Preferences (mark ‘X’ where applies)													
Yes							Vegan						No Pork							
No		X					Vegetarian						Kosher							
							Other						None					X		
Describe ‘Other’ Cultural/Religious Food Preferences																				
Food Allergies (mark ‘X’ where applies)																				
Gluten/Wheat				Fish/Shellfish								Peanut/Tree Nut								
Lactose Intolerant				Other								NKFA *					X			
Describe ‘Other’ Food Allergies, if Applicable																				
For Nutrition Operations personnel only (mark ‘X’ where applies)																				
Did patient report Fair or Poor ‘Appetite History’?																Yes				X
																No				
Did patient report Recent and Unintentional ‘Weight Loss History?’																Yes				X
																No				
If patient answered ‘Yes’ to both questions, refer patient to RD.																				
Name of 68M providing screening								Name of patient’s assigned RD												
68M’s name								Clinical RD 1												

* No known food allergies.

**Table 5 ijerph-18-07430-t005:** The field hospital staffing model is designed to identify staffing needs for nutrition services in a field hospital setting with 125–500 patient beds.

**Position**	**Shift**	**Working Hours**	**FTE** **(~125 Beds)**	**FTE** **(~250 Beds)**	**FTE** **(~375 Beds)**	**FTE** **(~500 Beds)**
Food Service Dietitian *	AM	6:30 a.m.–2:30 p.m.	1	1	2	2
Food Service Dietitian *	PM	11:00 a.m.–7:00 p.m.	1	1	1	1
Clinical Floor Dietitian **	Day	8:00 a.m.–4:00 p.m.	0	1	2	5
Clinical Critical Care Dietitian ***	Day	8:00 a.m.–4:00 p.m.	-	-	1	2
On-Call Dietitian	PRN	As needed	1	1	1	2
Food Service and Clinical Nutrition Manager	Day	6:00 a.m.–6:00 p.m.	1	1	1	1
Nutrition and Dietetics Technicians, Registered (NDTR)	Day	12 h, with three breaks	4	8	13	20

* At a census level of 125 beds, the a.m. and p.m. food service dietitians can function as the clinical floor dietitians for patient care and production. ** A clinical floor dietitian is necessary at a census level of 250 beds to address nutrition consults and provide MNT to patients at-risk for malnutrition who are receiving care in the ICU. *** At census level of 375 patient beds, critical care dietitians become necessary to support active and consistent engagement with ICU patients and their medical team.

**Table 6 ijerph-18-07430-t006:** Example of the training program syllabus, designed to identify training requirements upon the day of arrival (day 1) and for early shift (a.m.) operations.

Training Preparation				
JNYMS Layout Orientation and Walk-Through of Service Areas			Read Food Service and Clinical Operations Standard Operating Procedures	
Day 1 Schedule				
Time	Nutrition Operations	Food Service Operations		Clinical Nutrition Operations
06:30–07:00	Arrive to JNYMS and report to command center; Attend morning brief with USPHS leaders.	Arrive to JNYMS and report to food service operations; Review Patient Administration Division tracker to identify newly admitted, transferred or discharged patients from JNYMS; Review ‘Diet Order Change and Supplement Request Form’ from night shift; Update ‘Patient Census and Diet Order Tracker’ and ‘Diet Rosters.’		Arrive to JNYMS and report to clinical nutrition operations; Review Patient Administration Division tracker to identify newly admitted patients and current ICU census.
07:00–07:30		Shadow 68Ms while they verify ‘Diet Rosters’; Update ‘Patient Census and Diet Order Tracker’ and ‘Diet Rosters’ no later than 07:30.		Review consulting process. Collect and triage ‘Nutrition Consultation Forms’ from the night shift.
07:30–08:00	Observe food service operations.	Observe and assist 68Ms build meal delivery carts and begin delivering meals.		Observe critical care nutrition team; Attend ICU rounds; Review of medical documentation process using the ‘Modified ADIME Note’; Review ‘Enteral Nutrition Formulary’ and ‘Enteral Nutrition Support Calculator.’
08:00–09:30		Patient breakfast meal hours (08:30–09:30); Review Patient Administration Division tracker to identify newly admitted, transferred or discharged patients from JNYMS; Update ‘Patient Census and Diet Order Tracker’ as needed.		
09:30–10:00		Tally meals delivered by diet order type referencing the ‘Diet Roster’; Confirm total meals delivered with end of meal patient census; Update ‘Patient Census and Diet Order Tracker’; Observe equipment sanitation procedures as 68Ms sanitize meal delivery carts.		
10:00–10:30	Meet with food contractors.	Review Patient Administration Division tracker; Update ‘Patient Census and Diet Order Tracker’ and ‘Diet Rosters’; Shadow 68Ms while they verify ‘Diet Rosters’ on assigned wards; Update ‘Patient Census and Diet Order Tracker’ and ‘Diet Rosters’ no later than 10:30.		
10:30–11:00		Observe and assist 68Ms build meal delivery carts and begin delivering meals.		
11:00–12:00	Observe food service operations.	Patient lunch meal hours (1100–1230); Routinely monitor Patient Administration Division tracker for newly admitted patients; Update ‘Patient Census and Diet Order Tracker’ as needed.		RD Lunch Break.
12:00–12:30				Shadow floor nutrition team in the ICU; Observe 68Ms while they conduct nutrition screening using the ‘Nutrition Screening Form’; Perform medical documentation using the ‘Modified ADIME Note’; Shift change briefings with late shift clinical RDs.
12:30–13:00		Tally meals delivered by diet order type referencing the ‘Diet Roster’; Confirm total meals delivered with end of meal patient census; Update ‘Patient Census and Diet Order Tracker’; Observe equipment sanitation procedures as 68Ms sanitize meal delivery carts;		
13:00–14:30	RD lunch and shift change briefings with late shift nutrition ops RD.	RD lunch break and shift change briefings with late shift food service RDs.		

Note: In order to maintain wellness of the JNYMS staff, breaks are coordinated amongst the teams and are not outlined in this schedule. All staff are strongly encouraged to take breaks as needed being mindful of eating/drinking constraints due to PPE while in JNYMS. Practice proper don/doff procedures per the field hospital’s infection control team instructions.

## Data Availability

Not applicable.

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
