# Peer review of "Providing Food and Nutrition Services during the COVID-19 Surge at the Javits New York Medical Station"

_ijerph, 2021, doi:10.3390/ijerph18147430_

Round 1

Reviewer 1 Report

The reviewed article is not of a research nature, it is a case study describing part of the reality of medical management and showing the problematic fragments of the activity of the described entity. The presented case study describes the current conditions related to the situation of a fragment of the public health management system. The authors have comprehensively but synthetically characterized the Javits New York Medical Station (JNYMS) field hospital case. The main content of the case study and the described challenges is the issue of planning the diet and managing this process in extra-coordinated conditions, partially devoid of precise procedures. The developed standards of operation are presented as procedures, decision algorithms, service protocols and standardization of personnel training. The conclusion is the proposal to standardize the activities of the Nutrition Response Toolkit for Humanitarian Crises as operational protocols that can be implemented in an emergency.

I consider the article as well written and clearly showing the presented issues. The only reservations concern the relatively limited range of readers' interest. The article refers to American conditions. The authors did not analyze the possibility of implementing the described activities in other countries with different specificity and different standards of public health management. As a case study, however, it is an interesting text, undoubtedly broadening the spectrum of knowledge by describing the experiences of a subject forced to function in special and specific conditions.

The language of the manuscript is clear. The structure does not raise any objections. Additional visual and source materials add value to the article.

Reviewer 2 Report

This paper represents a relevant contribution to the literature related to food and nutrition services during the COVID-19 surge at the Javits New York Medical Station. Context itself seems reasonable, and certainly opens an avenue for discussion. The introduction to the paper nicely suggests that there still are gaps in knowledge that can be filled with the type of research conducted in this study. However, in my opinion some further considerations are the following:

  • I suggest considering a general, integrative theoretical approach to present a conceptual framework previously, and then write the paper from the angle of the specific chosen approach. The authors should tell summarize the theory they developed in the paper and tell readers why the theory they developed in the paper is important.
  • The paper should be better connected to the literature especially the gaps and contradictions in existing literature. The authors need to provide a much deeper review of literature in this field and explain how their research will help in understanding these problems. Moreover, some paragraphs regarding Introduction, Overview of JNYMS Structure, Staffing, and Timeline, and JNYMS NuOp Mandates should be well developed in this manuscript.
  • Even though the paper is empirically oriented, good papers (either theoretical or empirical) always provide a review of both types of papers related to their topic of study. The reasons for this are that different types of readers may be interested in reading the paper and more importantly it helps to better evaluate the merits of the paper’s contribution.
  • In my opinion, results are more indicative rather than representative. Limitations and managerial contributions of the study in terms of the generalization of the findings should be added.
  • The conclusion section is too brief and requires more elaboration in previous sections. Particularly, a Conclusion section regarding the ways in which this research with differing intervention contributes to managerial and theoretical implications in the study is required.

Reviewer 3 Report

The manuscript addresses an important topic of the challenges in providing food and nutrition services in JNYMS during the COVID-19 surge and provides solutions. The significance of this manuscript is obvious.

I have one suggestion. So far this manuscript is completely a case study, meaning that the authors did not compare the challenges faced by JNYMS and other similar facilities. If the problems faced by JNYMS are not just specific to JNYMS and are shared in other facilities, the toolkit developed by the authors will be more valuable and important. So the authors should consider adding a section discussing common challenges faced by similar facilities, which can provide a bigger picture to the readers and also make the contribution of this manuscript more significant.

Round 2

Reviewer 2 Report

The authors have made some improvements compared to the initial submission. However, in my opinion there are some considerations that limit the field of observation:

  • In terms of explaining the paper contribution from theoretical point of view, the authors should improve some sections to state more related references related to their work.
  • I find it a little confusing to read in the Introduction Section, some realities that have been used in this paper. I suggest that these considerations could be included according not only in the Introduction, but also in ‘2. Overview of JNYMS Structure, Staffing, and Timeline’ section.
  • The methodology needs more justification in the ‘3. JNYMS NuOp Mandates’. Additionally, data section should be clearly described in this case study, that is, several tables to establish some validity, and reliability are missing.
  • The Conclusion section is too brief. Please, develop your conclusions and recommendations.

Author Response

Thank you for your response regarding the manuscript entitled “Providing food and nutrition services during the COVID-19 surge at the Javits New York Medical Station.We appreciate the time and effort dedicated to providing feedback on our manuscript.Please see below, in blue, for a point-by-point response to the provided comments and concerns.

Reviewers’ Comments to the Authors:Reviewer #2

The authors have made some improvements compared to the initial submission. However, in my opinion there are some considerations that limit the field of observation:

Author response: Thank you for noting some of the improvements made to the initial submission

In terms of explaining the paper contribution from theoretical point of view, the authors should improve some sections to state more related references related to their work.

Author response: This request is very vague. We had provided over 60 references that support and justify our work.

I find it a little confusing to read in the Introduction Section, some realities that have been used in this paper. I suggest that these considerations could be included according not only in the Introduction, but also in ‘2. Overview of JNYMS Structure, Staffing, and Timeline’ section.

Author response: This request is very vague. It is not clear which considerations and realities the Reviewer refers to. Each section of this paper is based on real, lived, and detailed experience of the team that built the whole food delivery operation in extreme conditions of the high risk of exposure to disease to which we still have no cure, and the high time pressure because hospitals and ICUs were overloaded, and the solution must be found.

The methodology needs more justification in the ‘3. JNYMS NuOp Mandates’. Additionally, data section should be clearly described in this case study, that is, several tables to establish some validity, and reliability are missing

Author response: This request is quite misguided. We are presenting a case study and are operating under a well-accepted guidelines of presenting in-depth, multi-faceted explorations of complex issues in their real-life settings (see: Crowe, S., Cresswell, K., Robertson, A.et al.The case study approach.BMC Med Res Methodol11,100 (2011). https://doi.org/10.1186/1471-2288-11-100). We described both the process of learning about the case and the product of our learning, which is the Nutrition Response Toolkit for Humanitarian Crises, aiming to provide grounded recommendations --based on state-of-art knowledge in nutrition --designed to assist nutrition personnel in conducting nutrition services in a variety of emergency settings.

In presenting our study we applied the structure that has been utilized by another case study published in this Special Issue https://doi.org/10.3390/ijerph17238976(see ref 56 in our paper). The request to establish validity and reliability is unsubstantiated and it is notclear in what context reviewer use these terms.

The Conclusion section is too brief. Please, develop your conclusions and recommendations.

Author response: The conclusion provides a clear and consisted summary of the paper. The recommendations have been provided for each specific key element of the case study. For example, Table 1 illustrates examples of food served. Table 2 shows an example of diet roster. Table 3 shows an example of meal distribution given existing hospital capacity. Table 5 shows an example of nutrition screening form. Table 6 illustrates a scheme for staffing needs. Table 7 contains items for training. All presented tables serve as recommendations and the ready-to-use templates are detailed in the Nutrition Response Toolkit for Humanitarian Crises. The Discussion clearly outlines the recommendations and directions for training and for developing further research in part motivated by the prior set of comments offered by Reviewer 2.

Senior Author response: As the senior author, I take full responsibility for the suggested structure of the paper and framing recommendations and conclusions. As a researcher with near 40 years of experience, I published over 200 papers that were primarily stemming from original research, theoretical assumptions. I truly believe in the high value of well-documented experiences and practical recommendations calling for high transparency in sharing knowledge, even if such experience is not derived from a scientifically planned experiment with the solid theoretical foundations for a design and a proposed intervention. I have first-hand experience when data, concepts, and solutions emerging from a crisis were simply lost. And each time a crisis struck we operate like we did not learn our lessons from the past. We recognized that the presented experience could not be fully reproduced we don’t wish anyone to experience any crisis of such magnitude as the ongoing pandemic. Yet, we also recognize that emergencies are happening and the best response to emergencies is to be prepared, to reflect on what we learn, and pass this knowledge to many. The collective experience will determine which theoretical point of view survives.

Reviewer 3 Report

I have no further questions on my part.

Author Response

I have no further questions on my part.

Author response:Thank you for the very much for the comment.On behalf of co-authors, Ms. Gelfand, Mr. Perkins, Ms. Tarnas, Mr. Simpson, Mr. McGee, and Dr. Naumova I thank you in advance for your time and consideration